# Impacts of Climate Change on the Biogeography of Three Amnesic Shellfish Toxin Producing Diatom Species

**DOI:** 10.3390/toxins15010009

**Published:** 2022-12-22

**Authors:** Francisco O. Borges, Vanessa M. Lopes, Catarina Frazão Santos, Pedro Reis Costa, Rui Rosa

**Affiliations:** 1MARE—Marine and Environmental Sciences Centre & ARNET—Aquatic Research Network, Faculdade de Ciências, Universidade de Lisboa, 1749-016 Lisboa, Portugal; 2Departamento de Biologia Animal, Faculdade de Ciências, Universidade de Lisboa, 1749-016 Lisboa, Portugal; 3IPMA—Portuguese Institute for the Sea and Atmosphere, 1749-077 Lisboa, Portugal; 4S2AQUA—Collaborative Laboratory, Association for a Sustainable and Smart Aquaculture, 8700-194 Olhão, Portugal; 5CCMAR—Centre of Marine Sciences, Campus de Gambelas, University of Algarve, 8005-139 Faro, Portugal

**Keywords:** biogeography, climate change, species distribution models, harmful algal blooms, amnesic shellfish poisoning

## Abstract

Harmful algal blooms (HABs) are considered one of the main risks for marine ecosystems and human health worldwide. Climate change is projected to induce significant changes in species geographic distribution, and, in this sense, it is paramount to accurately predict how it will affect toxin-producing microalgae. In this context, the present study was intended to project the potential biogeographical changes in habitat suitability and occurrence distribution of three key amnesic shellfish toxin (AST)—producing diatom species (i.e., *Pseudo-nitzschia australis*, *P. seriata*, and *P. fraudulenta*) under four different climate change scenarios (i.e., RCP-2.6, 4.5, 6.0, and 8.5) up to 2050 and 2100. For this purpose, we applied species distribution models (SDMs) using four abiotic predictors (i.e., sea surface temperature, salinity, current velocity, and bathymetry) in a MaxEnt framework. Overall, considerable contraction and potential extirpation were projected for all species at lower latitudes together with projected poleward expansions into higher latitudes, mainly in the northern hemisphere. The present study aims to contribute to the knowledge on the impacts of climate change on the biogeography of toxin-producing microalgae species while at the same time advising the correct environmental management of coastal habitats and ecosystems.

## 1. Introduction

The continuing accumulation of greenhouse gases in the atmosphere (e.g., carbon dioxide—CO_2_) over the past centuries has led to significant changes in the global ocean [1,2]. Among these, changing oceanic chemistry, rising sea surface temperatures, and shifting oceanic currents are set to yield a vast array of negative impacts on marine ecosystems worldwide [3,4]. The past few decades have revealed an increasing understanding of the potential for marine climate change to significantly impact the frequency, magnitude, and geographical extent of harmful algal blooms (HAB) [5,6,7]. These phenomena have gained a particular degree of notoriety in the past few years due to their role in a wide variety of environmental issues [8,9]. Indeed, when environmental conditions are conducive to algal growth [10,11], there is the potential for the exacerbated population growth of marine and freshwater phytoplankton, with a high risk of significantly impacting ecosystems and human health [12,13]. Some of the more concerning impacts of HABs include mass die-offs of fish and shellfish [14,15] and the deaths of marine mammals and seabirds [16,17], which could lead to the collapse of coastal ecosystem community structures [18]. At the same time, HABs are also linked to outbreaks of human shellfish and finfish poisoning, which severely threaten human health [11,19]. Approximately 6% of the existing microalgae species (i.e., circa 300 out of 5000) have the potential to give rise to HAB events under favorable conditions [12,20]. From these, more than one hundred marine phytoplanktonic species can produce toxins with varying degrees of toxicity for marine species and other organisms in the food web [13,21] and have been linked to significant harmful impacts on marine communities and ecosystems [22]. 

One of the most serious HAB-related shellfish poisoning syndromes—amnesic shellfish poisoning (ASP)—is caused by domoic acid (DA) [23]. This potent neurotoxin is produced by diatom species (Bacillariophyceae) of the genera *Pseudo-nitzschia* and *Nitzschia* and can be accumulated by a wide variety of shellfish species—from mollusks to crustaceans—and some fish species [24,25], posing a severe risk for the health of animals and humans [21]. Indeed, DA is known to have widespread human health effects [26], negative impacts at various levels within food webs, and severe economic consequences for molluscan shellfish harvesters [27]. The genus *Pseudo-nitzschia* is distributed globally, occurring in both warm and cold climates, with a particular abundance in coastal areas [27,28]. Approximately twenty-six species are known to produce DA, including *P. australis*, *P. seriata*, and *P. fraudulenta*. Changing oceanic conditions are known to have contributed to the increased frequency and range of HABs in coastal areas worldwide [29]. Indeed, ocean warming and increased CO_2_ availability are known to benefit HAB species compared to other microalgae species [30,31]. At the same time, northward shifts for HAB species have been observed in the Northeast Atlantic [32,33,34], which constitute potential poleward migrations with progressive warming [6]. This is the case for species of *Pseudo-nitzschia* spp., with new blooms occurring in previously unseen areas [35] and extreme events triggering intense HAB events [36,37]. 

Given the high likelihood of HAB events to increase alongside CO_2_, temperatures, and changing climate patterns [6], it is paramount to model potential changes in the distribution of HAB species. One way to predict potential changes in the distribution of HAB species under marine climate change is to employ species distribution models (SDMs). In short, SDMs combine the biogeographic knowledge in climate change impact studies and modeling, allowing the projection of the impacts of future environmental change on biodiversity and ecosystem health [38,39,40]. These models take georeferenced occurrence data for a given species or set of species, together with environmental and abiotic predictors for a defined geographical extent, establishing a relationship between them and defining a species’ ecological niche. This, in turn, allows researchers to project potential changes in distribution ranges across time and/or space [41]. Although being bounded by a set of assumptions and limitations which must be considered [42,43,44,45], these models offer a suitable framework for predicting changes in species distributions and are applicable to large species assemblages and across vast geographical and temporal spaces [46]. For this reason, SDMs have observed a steep rise in their development and use in the past few decades [47,48] and are increasingly being used to project the effects of climate change on the distributions of species and communities [49,50,51,52]. 

Not surprisingly, these models have already been employed to project potential marine climate change induced impacts in the distribution of phytoplankton [30], especially for HAB species [5,53,54]. In this context, the objective of the present study was to evaluate the potential effects of future marine climate change on the distribution of three amnesic shellfish toxin (AST)-producing *Pseudo-nitzschia* species: (i) *P. australis*; (ii) *P. seriata*; and (iii) *P. fraudulenta*. To achieve this, this study implemented an SDM workflow using MaxEnt modeling to model the present-day habitat suitability and species distribution, projecting these into two future time periods (i.e., 2050 and 2100) and four representative concentration pathways (RCP; RCP-2.6, -4.5, -6.0, and -8.5; CMIP5) and quantifying spatiotemporal trends. 

## 2. Results

### 2.1. Variable Contributions

Regarding the variable contributions in the ensemble models, the bathymetry predictor was consistently the most important variable across all three species, at over 60% (Table 1), while the dynamic environmental variables contributed considerably less (i.e., 15% and under). Specifically, the next three most important variables for the three species were temperature-related layers, apart from *Pseudo-nitzschia fraudulenta*, in which the salinity minimum was the second most important predictor, at 15.2%. 

### 2.2. General Patterns of Habitat Suitability

Concerning the projected changes in the average distribution, a common trend was found between the three species. First, the latitudinal distribution centroids for all species were predicted to occur in the northern hemisphere. Then, concerning the projected changes, all species exhibited a northward shift in their centroid of latitudinal distribution until the middle of the century (Figure 1). However, between 2050 and the year 2100, the species’ responses were different. For *P. australis*, the ensemble model projected a considerable northward movement of the species between the present day and the middle of the century (i.e., 2050), moving back southward for RCP-2.6 and 4.5, continuing its movement northward for RCP-6.0 (albeit to a considerably lesser extent) or stabilizing until the end of the century for RCP-8.5 (Figure 1A). Regarding *P. seriata*, this species exhibited a continuous northward movement between the present day and 2050 and until 2100, which was exacerbated with increasing RCP scenario severity, a trend that was particularly evident when comparing the species position by 2100 across RCP scenarios (Figure 1B). Finally, *P. fraudulenta* exhibited a clear expansion northward, followed by a contraction southward for most RCP scenarios, except for RCP-8.5 (Figure 1C), where the species continued its movement northward until the end of the century. 

### 2.3. Habitat Suitability and Occurrence Distribution: Latitudinal Trends

The following section holds the outputs for the latitudinal trends in mean habitat suitability for the three species and the projected differences in the binary occurrence distribution across time for each RCP scenario. Regarding *P. australis*, the mean latitudinal habitat suitability decreased over time for almost the entire latitudinal profile (Figure 2). This decreasing trend relative to the present day baseline not only increased in magnitude between 2050 and 2100 but was also exacerbated along the RCP scenarios, with RCP-6.0 and -8.5 exhibiting the greatest differences compared to the present day, mainly until the end of the century (Figure 2F,H). Very limited latitudinal bins were projected to experience increasing mean habitat suitability, which were mostly confined to the higher latitudes in the northern hemisphere, with the maximum values being projected for the 2100 horizon for RCP-4.5 and RCP-6.0 (Figure 2D,F). 

These changes were followed by dynamic trends in this species’ occurrence distribution over time (Figure 3). Indeed, for all RCP scenarios the areas of projected extirpation outnumbered those of projected expansion, a trend that was exacerbated by RCP scenario severity. Specifically, in RCP-2.6, the projected distribution loss was relatively limited, with major areas including some areas of the Atlantic Ocean (North Carolina, the Gulf of Mexico, and southern Brazil (by 2050) and the offshore Bay of Biscay and North Sea (by 2100)); Southeastern Pacific Ocean (Chilean seas (2100)); and the Bass Strait (by 2050 and 2100) in the Indian Ocean (Figure 3A). The larger areas of projected expansion for this species were located in some areas of the Northeastern Atlantic Ocean (by 2050 and 2100) and in the northern North Sea. Other localized expansions also occurred in the Iberian Peninsula (2050) and Morocco (2050), and lastly in northern New Zealand (2050). This scenario also predicted large areas of projected transitory fluctuations, namely areas of contraction followed by expansion in the Caribbean Sea and northern South America and on the Peruvian coastline and the north coast of Australia. Projected expansion followed by distribution contraction occurred, namely in the North Sea and on the African coastlines of the Benguela current. RCP-4.5 exhibited a relative exacerbation of these trends (Figure 3B). Indeed, most of the predicted areas of occurrence in the Western Atlantic exhibited extirpation by 2050 (Gulf of Mexico and Caribbean Sea). At the same time, localized losses also occurred on the northeastern Australian coastline and in the Bering Strait (2050). In this scenario, however, the Northeastern Atlantic Ocean and the North Sea did not exhibit distributional losses, with these regions exhibiting the largest gains until the end of the century. As in the previous scenario, localized gains also occurred along the northern and southern coastlines of the Iberian Peninsula (2050) and Morocco (2100) as well as sparsely along the eastern Australian and northern New Zealand coasts. For RCP-6.0, a smaller relative change in mean habitat suitability until 2050 (Figure 2E) compared to RCP-4.5 led to considerably fewer areas of projected extirpation occurring by 2050 (Figure 3C). However, a considerable decrease until 2100, more severe than that projected for RCP-4.5 (Figure 2F), resulted in a substantial surge in areas of extirpation by the end of the century. Indeed, extirpation was projected for most of the Western Atlantic Ocean and the Eastern Australian coastline, with localized losses also occurring in the North and Norwegian seas and on the coast of the tip of south America (i.e., southern Chile and Argentina). In this scenario, gains were again mostly limited to the Northeastern Atlantic Ocean (2050), parts of the North Sea (2100), the Iberian Peninsula (2050/2100), and Morocco (2100). Major areas of fluctuation included areas of expansion followed by contraction in 2100 on the North and Eastern Australian coastlines and in the North Sea and southern Argentina, while contraction followed by end-of-century expansion were projected for the Celt Sea. Lastly, in RCP-8.5 these changes were again aggravated, with most of the Eastern and Western Australian coasts exhibiting losses in distribution as well as most of the western Atlantic Ocean (Figure 3D). In this scenario, gains remained sparse in the Iberian Peninsula and Morocco, while again the Celt and North Seas and the Northern Atlantic Ocean exhibited the largest expansion areas until the end of the century. 

Regarding *P. seriata*, the mean latitudinal habitat suitability changes showed an overall decrease until the end of the century for most of the latitudinal profile (Figure 4), together with a considerable increase in suitability for the northernmost latitude bands (i.e., the subarctic and arctic latitudes). Habitat suitability for this species did not change, overall, for the southern hemisphere, despite a relative decrease south of the equator, which became more apparent with increasing RCP severity (e.g., see Figure 4B,H). In the northern hemisphere, however, subtemperate and temperate regions exhibited the greatest losses in habitat suitability until the year 2100, particularly in the two most extreme scenarios (Figure 4F,H). In the northernmost latitudes, habitat suitability could be seen to increase consistently alongside time and to a greater magnitude with increasing scenario severity. In terms of changes in occurrence distribution (Figure 5), there was a persistent pattern across RCP scenarios of distribution loss at lower latitudes and distribution gains at higher latitudes in the northern hemisphere. Changes in distribution were minimized for RCP-2.6 (Figure 5A). Indeed, in this scenario, areas of loss were very restricted in space and time, with some occurring in the northern area of the Black Sea (2050) and on the Canadian and Greenlandic shores of the Labrador Sea (2050). Areas of expansion included northern Canada (in Hudson Bay), in the Arctic Ocean (e.g., Svalbard and the Barents Sea), and in the Okhotsk Sea in the Pacific Ocean. For RCP-4.5 (Figure 5B), the species underwent a considerable distribution restriction in the Black Sea (2050) while also experiencing distribution loss in the Labrador Sea (2050), as in the previous scenario. In terms of gains, this species expanded again in the northernmost latitudes, namely in Hudson Bay, the Artic Ocean, and the Okhotsk Sea (2050 and 2100), and was projected to expand considerably into the northern Baltic Sea. In RCP-6.0, decreasing habitat suitability over most of the latitudinal profile led to a projected loss approaching the known species distribution from the south (Figure 5C). In its current range, however, *P. seriata* was projected to undergo more limited losses in the Black and Labrador seas compared to the previous scenarios. Indeed, along the Canadian and Greenlandic shores of the Labrador Sea, it exhibited oscillating habitat suitability between the present day, the year 2050, and 2100, leading to a projected contraction in this region until the middle of the century, followed by a new expansion until 2100. In this scenario, the species mostly expanded in the Bering and Okhotsk seas and in the Arctic Ocean, with limited expansion in the northern Baltic Sea and in Hudson Bay. Lastly, under RCP-8.5, the decreasing habitat suitability over most of the temperate latitudes of the northern hemisphere led to an encroaching loss of potential distribution (Figure 5D). However, gains were also maximized in RCP-8.5 in the same regions as in the previous scenarios, albeit with most occurring only by the end of the century. 

Finally, concerning *P. fraudulenta*, this species again exhibited decreasing habitat suitability over most of the latitudinal profile (Figure 6), except for the highest latitudes in both the northern and southern hemispheres. Decreasing mean habitat suitability was also maximized, in general, for the southern tropical and northern temperate latitudes, a trend that was exacerbated by increasing RCP severity (see Figure 6B,H). However, it is worth noting that the magnitude of suitability change did not increase over time for all scenarios. Indeed, for RCP-2.6 and RCP-6.0, changing habitat suitability was greater between the present day and the year 2050 than between the present day and the end of the century (Figure 6A–D). Regarding the distribution of *P. fraudulenta*, this species exhibited sparse areas of projected extirpation in most of the world’s oceans by 2050 (Figure 7A), namely some areas of the Mediterranean Sea (e.g., in the Adriatic and on the Spanish coast); in the Northeast Atlantic (e.g., the Celt Sea and north of Scotland); in the Indic Ocean (i.e., near Pakistan and India); in the Pacific Ocean (i.e., in the southeast of the Sea of Japan and in the Gulf of California); and finally in the Gulf of Mexico and in northern Australia. In terms of gains, these were again very limited for this scenario and were mainly located at the highest latitudes of both hemispheres (i.e., in the northwestern Atlantic and northeastern Pacific oceans, certain areas of the Arctic Ocean, and in southern areas offshore Argentina and Chile). For RCP-4.5, the projected areas of expansion remained similar, but there were increases in the areas of projected loss, concurrent with decreasing habitat suitability (Figure 7B). Indeed, the abovementioned areas of projected loss underwent projected growth until the end of the century for this scenario, mainly in the Arafura Sea in Northern Australia (2100) and in the Pacific and Indic oceans. Under RCP-6.0 and RCP-8.5, considerably larger decreases in habitat suitability led to considerably larger areas of extirpation in most of the world’s oceans, e.g., in the Western Pacific, in the Mediterranean Sea, and sparsely spread around the Atlantic Ocean (Figure 7C,D). In these scenarios, expansion was particularly evident in the Gulf of Alaska and the Arctic Ocean (e.g., in the Barents Sea and Svalbard) and near the southern tip of South America. 

## 3. Discussion

The present article aimed to conduct a global analysis of the biogeographic responses of three AST-producing diatom species to the effects of marine climate change until the end of the century, akin to previous work conducted on paralytical shellfish toxin (PST)-producing species [54]. Overall, the models were able to accurately predict the accepted present-day distribution of each *Pseudo-nitzschia* species (Table 2) [27,35], despite some instances of over- and underprediction (e.g., *P. seriata* had wide overprediction in the southern hemisphere), and common trends were found between the three AST-producing species. 

All species were projected to undergo considerable decreases in mean habitat suitability across most of the latitudinal profile, regarding the present day, leading to potential extirpations across (when the species is currently present) equatorial, tropical, and temperate regions. Moreover, despite decreasing along most of the latitudinal profile, the mean habitat suitability was seen to increase over time and across RCP scenarios for all species at higher latitudes (i.e., the polar and subpolar regions). These changes in habitat suitability resulted in specific distributional changes, which despite being different between species, shared considerable northward shifts in the distributional centroids between the present day and (at least) the middle of the century (followed by some relative oscillations for *P. australis* and *P. fraudulenta*) and the year 2100 (for *P. seriata*). These results hint at potential poleward distributional shifts until the end of the century for these *Pseudo-nitzschia* species. Concerning the variable contributions, bathymetry was the most important variable for all species, with values between approximately 61% (*P. fraudulenta*) and approximately 82% (*P. seriata*). This trend was also observed in SDMs with PST-producing dinoflagellates [54] and is to be expected when dealing with neritic species and with occurrence data mostly obtained from coastal monitoring programs. At the same time, *Pseudo-nitzschia* blooms are known to be linked to upwelling regions [55,56]. Afterwards, temperature layers were, overall, the most ecologically relevant predictors. Indeed, rising salinity has been shown to favor *Pseudo-nitzschia* spp. abundances [57], and as such, the salinity minimum held a significant contributive role for the models of *P. fraudulenta*.

Rising temperatures have been linked to poleward shifts in the thermal niches of phytoplankton until the end of the century, akin to a wide variety of marine taxa [58,59], suggesting sharp declines in tropical phytoplankton diversity in the absence of adaptation to warming [60]. Indeed, several prior studies projecting biogeographical climate change effects suggested similar phenomena could occur for phytoplankton species. Indeed, dinoflagellates in the Atlantic and Pacific oceans have been found to closely follow the rate of isotherm movement [10,61,62,63] as the range of the optimal environmental conditions shifts, effectively pressuring their ability to survive and adapt at lower latitudes and opening the possibility of HAB events in newer regions. Borges et al. [54] also projected potential poleward shifts for PST-producing dinoflagellate species (i.e., *Alexandrium minutum*; *A. catenella*; and *Gymnodinium catenatum*), while other groups such as Cocolitophores [64] also exhibited the same distributional changes. Changes in temperature have also been shown to induce significant impacts in diatoms, with warming being linked to lower cell yields and promoted growth in *Pseudo-nitzschia* species [65], while extreme events such as heatwaves have also triggered pelagic HAB events [37]. Indeed, a record-setting HAB event by *Pseudo-nitzschia* occurred as a direct consequence of a marine heatwave in the northeast Pacific Ocean between 2013 and 2015 [36]. Additionally, North Atlantic diatoms have been projected to shift the central positions of their core range poleward, leading to a significant reshuffling of the phytoplankton communities, with broad effects on food webs and biogeochemical cycles [30]. The present study presents similar results, particularly regarding the cold-water species *P. seriata*, which was projected to further restrict its southern limit of distribution in the northern hemisphere and consequently expand its range in the Arctic and subarctic regions. Despite exhibiting similar trends, *P. australis* and *P. fraudulenta* exhibited relatively smaller shifts north, expanding more limitedly in northern temperate and subarctic regions but also exhibiting significant restrictions at lower latitudes. Diatoms are known to have a lower activation energy compared to other phytoplankton groups [66], and tropical species have been shown to adapt to multigenerational exposure to warming through various thermal strategies, i.e., either by increasing the optimal growth temperature and maximum growth rate or by shifting from specialist to generalist, increasing the maximum critical thermal limit, albeit trading off photosynthetic efficiency, high irradiance stress, and a lower growth rate [67]. Despite their potential to adapt, the lower competitive fitness of diatom species induced by thermal stress could eventually lead to species extirpation at lower latitudes, where the ocean becomes too warm to support growth [39]. The concomitant migration of *Pseudo-nitzschia* into new ecosystems at higher latitudes, alongside their environmental thermal optimum, poses a significant risk to marine communities and to the human societies inhabiting these regions [6,35], with these new regions supporting a widening window of optimal conditions for blooms to develop [5,68]. However, since HAB events result from a complex interplay of abiotic and biotic interactions, the present results do not present a prediction of likelihood of ASP events. Instead, they contribute towards the projection of species’ movements and identifying potential areas at risk. 

Lastly, any modeling endeavor is required to address the existence of potential limitations in the analysis, which arise from methodological choices and assumptions. First, the models were susceptible to instances of over- and underprediction for the three species. Overprediction in SDMs is quite common, since the models assume that species will completely occupy areas that are calculated to be climatically suitable [42], ignoring other factors, such as unaccounted predictors, environmental variables, geographical barriers, etc., which prevent species occurrence at a local or regional scale [69]. In the present manuscript, the lack of readily available downscaled projections of environmental predictors such as nutrient ratios and light for the future and for each of the RCP scenarios that were employed presented a limitation to the ecological relevance of the models. Nutrient and light conditions are of paramount importance for shaping marine phytoplankton abundance and distribution, particularly for *Pseudo-nitzschia* spp. blooms [55,56,57]. In this sense, the ecological modeling of future marine climate change biodiversity impacts requires the expansion of already existing environmental and abiotic layer online databases (such as Bio-ORACLE, MARSPEC, etc.) so that these types of layers are readily available at global scales whenever possible. Concurrently, failing to predict the species presence in areas with existing historical records (i.e., underprediction) is directly linked to the nature of the occurrence point data. These data were mainly retrieved from online databases such as GBIF and OBIS (i.e., the Ocean and Biodiversity Information System), which do not provide a full inventory of a species’ known distribution, mainly due to geographical biases associated with sampling efforts due to accessibility and funding [70]. Nonetheless, instances of over or underprediction, given that they are not dominant in the model predictions, do not diminish the analysis’s predictive potential in suggesting the global trends of future poleward shifts or major extirpations of species’ distributions. Moreover, long-term SDM projections do not account for seasonal changes in the abundance of each species, which directly impact the existence of an algal bloom. Lastly, single-trophic-level SDMs do not incorporate the contributions of other trophic levels into the potential expansion or restriction in distribution induced by, for instance, predator species. Indeed, phytoplankton are vulnerable to strong predation pressure by many protozoan and metazoan grazers [71], which has been shown to influence toxin production in toxic species (e.g., *P. seriata*) and *Pseudo-nitzschia* species that were previously considered nontoxic (e.g., *P. obtusa*) [72,73]. As such, including these types of trophic interactions in the models by incorporating the projected distributions of relevant grazer species under the same scenarios could lead to more accurate estimates [74,75].

## 4. Conclusions

The present manuscript suggests an overall decrease in habitat suitability at lower latitudes at a global scale, followed by changes in the distribution ranges of the three modeled species into areas where they previously did not occur or only occurred locally. Indeed, for all species, poleward distribution shifts were projected by the ensemble models. These shifts pose serious ecosystem risks, opening the possibility of nuisance blooms in novel areas [76]. The inclusion of ecological modeling predictions and projections for marine species in management action, such as marine spatial planning, fisheries, aquaculture, and even coastal development, is of paramount importance when crossed with expert knowledge, as they allow for an accurate overview of potential future scenarios that might threaten ecosystem health and ecosystem service provisioning.

## 5. Materials and Methods

### 5.1. Data Collection and Curation

Data on each species’ occurrence were collected using the ‘OccurrenceCollection’ function of the ‘megaSDM’ package. This function, in short, collects occurrence data from the Global Biodiversity Facility (GBIF) database [77], with some degree of data precuration [78]. GBIF is an online network that links diverse data sources to compile and provide biodiversity data. To restrict the data retrieved in each dataset, a common set of filters was used when retrieving the occurrence data, specifically selecting “Human Observation” and “Preserved Specimen” as the bases of records and georeferenced data. At the same time, data featuring duplicate observations, improper datum conversion points, missing latitude or longitude values, and rounded coordinates were filtered out. The resulting compiled dataset was then curated using RStudio [79]. Since SDMs must ideally restrict model calibration to only accessible areas, all occurrences were restricted by a maximum depth limit of approximately 200 m, which is the accepted value for the mean maximum depth of the continental shelf [80]. To achieve this, the occurrence data for each species were converted into spatial polygon objects, which were used to extract the depth values at each point coordinate from a bathymetry raster layer (at a resolution of 5 arcmin) obtained from ocean climate layers for marine spatial ecology (MARSPEC) [81]. This was performed using the function ‘extract’ from the package ‘raster’. Afterwards, the spatial polygon objects were converted into the data frame format and merged with the georeferenced depths, and the new data frames were divided to exclude depths greater than 200 m [5,54]. A second restriction clipping was employed to remove all points based on land. To achieve this, a shapefile from the world’s ocean bodies (Natural Earth Data, https://www.naturalearthdata.com (accessed on 14 February 2022)) was used to remove spatial points outside the world’s oceans. Table 3 presents the postcuration number of valid entries per species. The curated occurrence dataset, the plotted dataset occurrences, and the R script used in this section are presented in the Appendix A.

### 5.2. Environmental Predictors

The present study used a total of four main predictor variables, specifically three oceanographic predictors: the sea surface temperature (SST), salinity, and current velocity (each including the global mean, maximum, minimum, and range layers) and one topographic variable (i.e., bathymetry). The predictor choice was based primarily on the availability of both present layers (i.e., 2000–2014 averages) and future projections (i.e., 2040–2050 and 2090–2100) for the four RCP scenarios of interest (i.e., RCP-2.6, -4.5, -6.0, and -8.5; CMIP5). These four RCP scenarios present different climate forcing possibilities, depending on the cumulative greenhouse gas emission trends until the end of the century. The scenarios employed in this article belong to the fifth phase of the Coupled Model Intercomparison Project (CMIP5), which is one of the latest climatic model phases that features downscaled layers. Specifically, the most optimistic scenario (i.e., RCP-2.6) requires a continuous decline in CO_2_ emissions between 2020 and 2100, projecting global temperatures to rise by less than two degrees by the end of the century. In RCP-4.5, the intermediate scenario, emissions would peak by 2045, resulting in a temperature increase of between 2 and 3 °C. The high-greenhouse-gas scenario (i.e., RCP-6.0), would result in a potential increase of 3 to 4 °C, while the “worst case scenario”, RCP-8.5, is based on continuously increasing emissions, leading to an increase of over 4 °C [82,83]. The layers for the three oceanographic variables were obtained from Bio-ORACLE, which offers global geophysical, biotic, and climate layers at a common spatial resolution (i.e., 5 arcmin) and a uniform landmask [84,85]. In turn, the bathymetry layer was retrieved from MARSPEC, as described in the previous subsection. 

### 5.3. Premodeling Procedures

A series of premodeling procedures was conducted using the package ‘megaSDM’ [78], which was also the main modeling package used in the analysis. Since ‘megaSDM’ relies on MaxEnt modeling, there was a need to reproject all predictor variables to an equal-area projection (i.e., the cylindrical equal-area projection: “+proj = cea + lat_ts = 0 + lon_0 = 0 + x_0 = 0 + y_0 = 0 + datum = WGS84 + no_defs”) using the method of the nearest neighbor. This was performed since MaxEnt randomly samples grid cells from the available geographic space, implicitly assuming cells of equal area in the entire extent of a given layer [86], which would inadvertently introduce sampling bias since conventional nonequal area projections have grid cells that vary in their areas when moving away from the equator. Afterwards, the functions ‘TrainStudyEnv’ and ‘PredictEnv’ were used to standardize the present and future input environmental data by clipping and resampling the raster predictors while also defining the training area where the occurrence and background points were located and the study area where the model would be tested [78]. 

To deal with the inherent bias of occurrence data collected from online databases [87,88], which decreases the overall accuracy of SDMs [87,89,90], the ‘megaSDM’ package employs environmental filtering of the occurrence data [90]. This procedure mitigates environmental and spatial biases by dividing the environmental data into a set number of bins (n = 25 in the present analysis) and selecting one point from each unique combination of bins to create a subset of occurrence points that have been filtered by the environment [78]. In short, this method allows for the removal of oversampled or clustered occurrence records, all the while maintaining the range of environments in which a species was found [90]. The number of post-environmental-filtering occurrences used in the present analysis is presented in Table 2. The script for the SDM analysis is presented in the Appendix A (see “sdm_analysis.R”). 

Since the collected occurrence data only featured presence data and SDMs must account for absence data by either incorporating true data or pseudo-absence data [91], a set of background points (n = 1000 per species) describing the environmental conditions of the training area were created for each species. The background points were created using a ‘combined’ method, which samples points both randomly and with a spatially constrained method [78]. In short, 50% of the background points were randomly sampled from the entire study area [92], while the other half were sampled from within buffers created around each true presence point. In the latter, the radius of each buffer was proportional to the 95% quantile of the distance to the nearest neighbor for each point [78]. This combined generation method has been shown to allow for a reduction in the model environmental suitability overestimation in regions with greater occurrence point density, which can occur in more easily sampled areas and constitutes a type of spatial bias [93,94]. At the same time, this method also reduces the susceptibility to errors of extreme extrapolation and overfitting induced by using purely spatially constrained methods [95]. The background points were also subjected to the process of environmental filtering, creating an even spread across the available environmental space while still retaining their spatial weighting [78]. 

### 5.4. Modeling

The ‘megaSDM’ package uses MaxEnt modeling to calculate the present habitat suitability and distribution of a given species [78]. The use of MaxEnt instead of several different modeling algorithms aims to prevent the introduction of uncertainty in the consensus model [95]. This technique employs maximum entropy methods and machine learning and is known to perform particularly well with presence-only species records [96] while maintaining a competitive predicted performance when compared with the highest-performing methods [95,96]. Moreover, this package allows for replication with subsequent ensembles to generate statistically rigorous models [78]. A replicate number of 5 was employed, meaning that the MaxEnt algorithm ran five times per species, each time with a different subset of occurrence points.

The function ‘MaxEntProj’ was used to evaluate each model replicate and the final ensemble. In short, this function compared the area under the curve (AUC) values for each model replicate, comparing them to null models where the multiple occurrence point replicates were placed randomly across the training area and to a random subset of the null model [78,97]. Afterwards, the function removed all models with a validation AUC lower than 0.70. The evaluation plots and tables for each species are presented in the Appendix A (see the “Evaluation” folder). The ‘MaxEntProj’ function was then used to project all models onto the current and future environments and across all four RCP scenarios. Subsequently, the median value of each pixel was calculated to create the ensemble raster of all replicate maps, thus reducing the potential effects of any existing outliers [78,98]. Binary maps of probability of occurrence (0 or 1) were obtained by employing a threshold value, specifically the mean model TSS criteria of model evaluation, the ‘maximum test sensitivity and specificity’ logistic threshold, which maximizes the specificity and sensitivity of the receiver operating curve (ROC) and is particularly effective for presence-only data [99] on continuous habitat suitability maps. The ensembles of habitat suitability for each species, time, and RCP scenario as well as the respective binary maps of probability of occurrence are supplied in the Appendix A (see the “Projections” folder).

### 5.5. Postanalysis

To investigate any potential changes in habitat suitability and species distribution across time and space, a series of visual and quantitative postanalysis procedures were employed. These procedures were the same as in Borges et al. [54], and the R scripts that were employed can be found in the Appendix A (see the “Post_analysis” folder). First, changes in the global latitudinal distribution were assessed by plotting the latitudinal centroid (i.e., the arithmetic mean latitude for the species-occupied cells) for each time and scenario. Second, the latitudinal trends in habitat suitability were calculated by converting each habitat suitability ensemble into a matrix, from which the mean value of each row (i.e., the latitudinal band) was calculated, resulting in a vector of mean habitat suitability across the latitudinal profile. The present-day vector was subtracted from the future projection vectors (i.e., 2050 and 2100), obtaining a vector of the changes in mean habitat suitability by latitude, which was plotted to visualize potential changes across time and RCP scenarios [78]. Then, changes in distribution were assessed by processing the binary maps of the probability of occurrence using the ‘createTimeMaps’ function of the ‘megaSDM’ package. This function generates maps that incorporate information on both projected unidirectional range shifts (i.e., range contractions or expansions) and transitory fluctuations (i.e., range contractions followed by expansions or vice versa) [78,100]. To achieve this, the function subtracts the projected future distribution of a given species from its predicted present-day distribution, generating a map of distributional change across time. The time maps for each species are supplied in the Appendix A (see the “Projections” folder). 

## Figures and Tables

**Figure 1 toxins-15-00009-f001:**
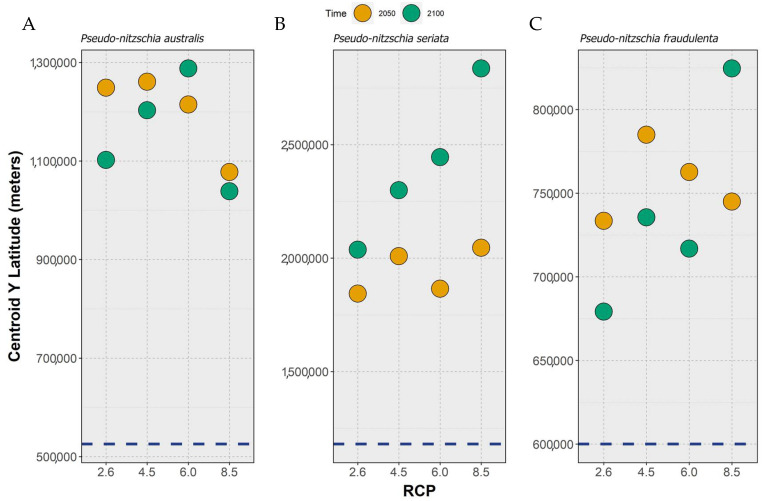
Changes in the position of the centroid of latitudinal distribution (i.e., the mean latitude of the occupied cells) across time and RCP scenarios (i.e., RCP-2.6, -4.5, -6.0, and -8.5; CMIP5) for (**A**) *Pseudo-nitzschia australis*, (**B**) *P. seriata*, and (**C**) *P. fraudulenta*. Positions calculated for 2050 (orange circles) and 2100 (green circles). The dark blue, horizontal dashed line represents the centroid position for the present day (i.e., 2000–2014, based on monthly averages).

**Figure 2 toxins-15-00009-f002:**
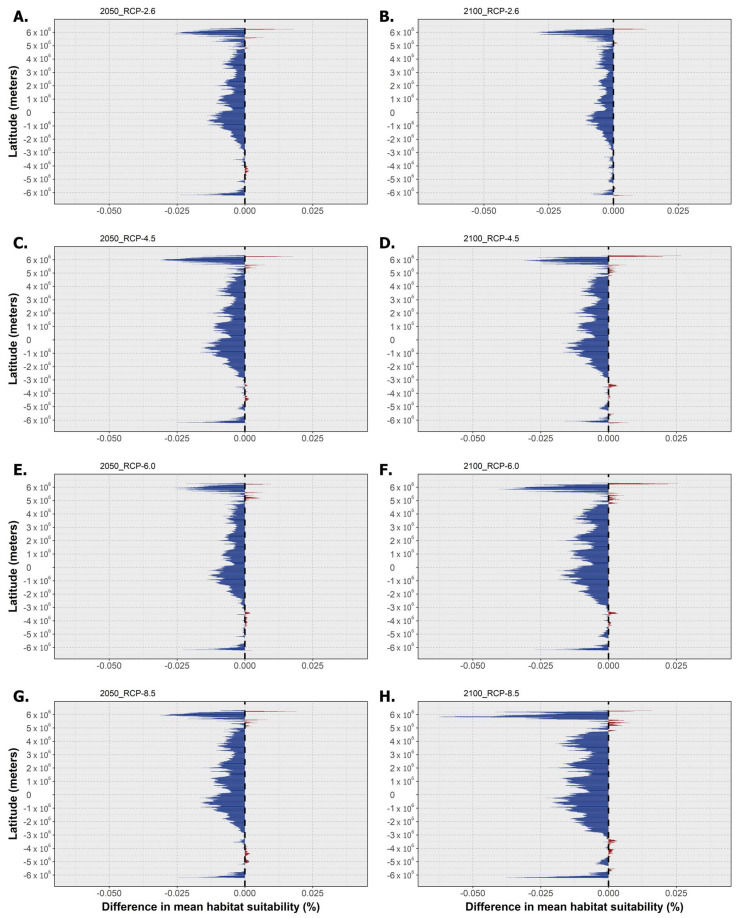
Mean latitudinal habitat suitability temporal changes (i.e., gains in red and losses in blue) between the present day and 2050 (left—(**A**,**C**,**E**,**G**)) and 2100 (right—(**B**,**D**,**F**,**H**)) for *Pseudo-nitzschia australis* across the four representative concentration pathway scenarios (RCP-2.6, -4.5, -6.0, and -8.5; CMIP5). The vertical dashed line at ‘0′ represents the present-day baseline.

**Figure 3 toxins-15-00009-f003:**
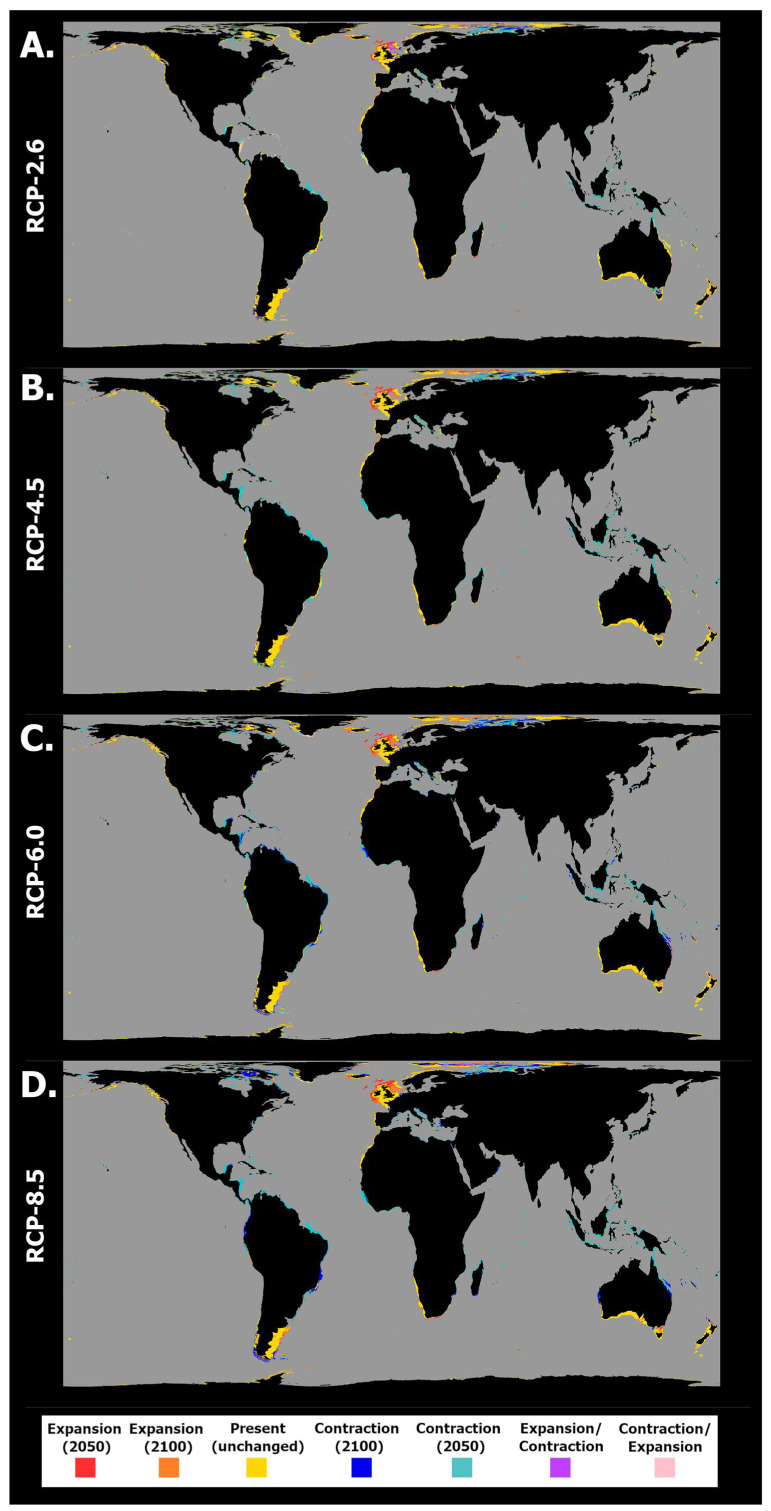
Projected spatiotemporal changes in distribution for *Pseudo-nitzschia australis* between the present day, 2050, and 2100 across four representative concentration pathway scenarios: (**A**) RCP-2.6, (**B**) RCP-4.5, (**C**) RCP-6.0, and (**D**) RCP-8.5. Projected occurrence distribution changes are represented: unidirectional range shifts (i.e., projected expansions in red and orange and projected contractions in dark and light blue) and transitory fluctuations (i.e., range contraction followed by expansion in pink and vice versa in purple).

**Figure 4 toxins-15-00009-f004:**
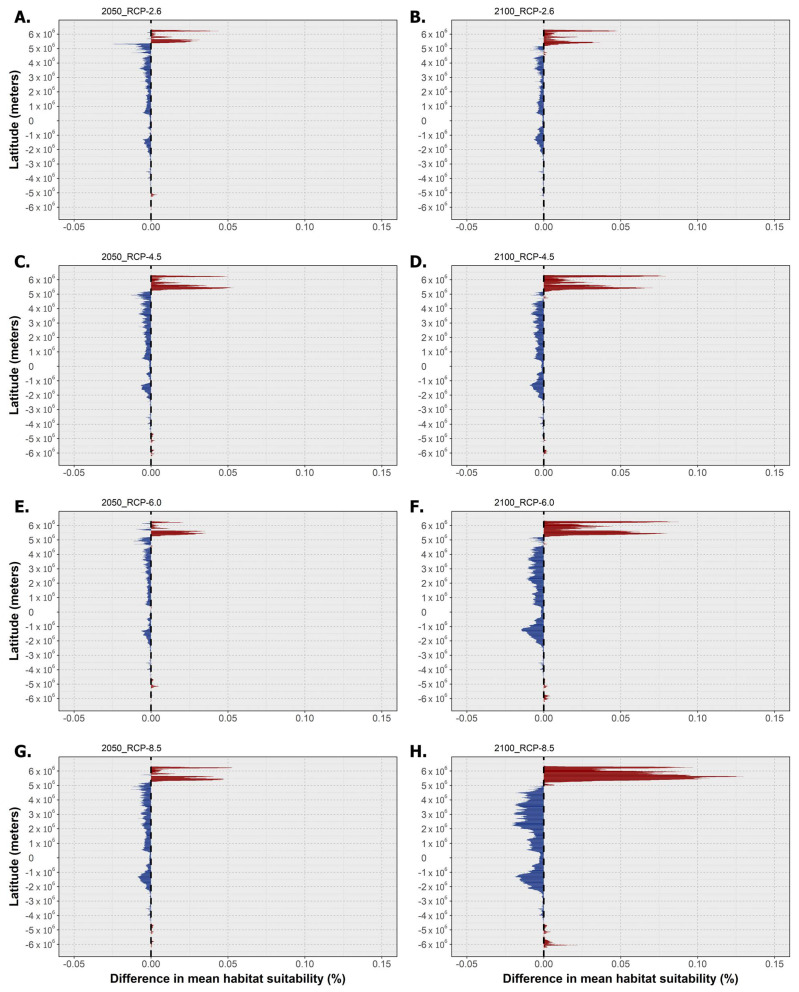
Mean latitudinal habitat suitability temporal changes (i.e., gains in red and losses in blue) between the present day and 2050 (left—(**A**,**C**,**E**,**G**)) and 2100 (right—(**B**,**D**,**F**,**H**)) for *Pseudo-nitzschia seriata* across the four representative concentration pathway scenarios (RCP-2.6, -4.5, -6.0, and -8.5; CMIP5). The vertical dashed line at ‘0′ represents the present-day baseline.

**Figure 5 toxins-15-00009-f005:**
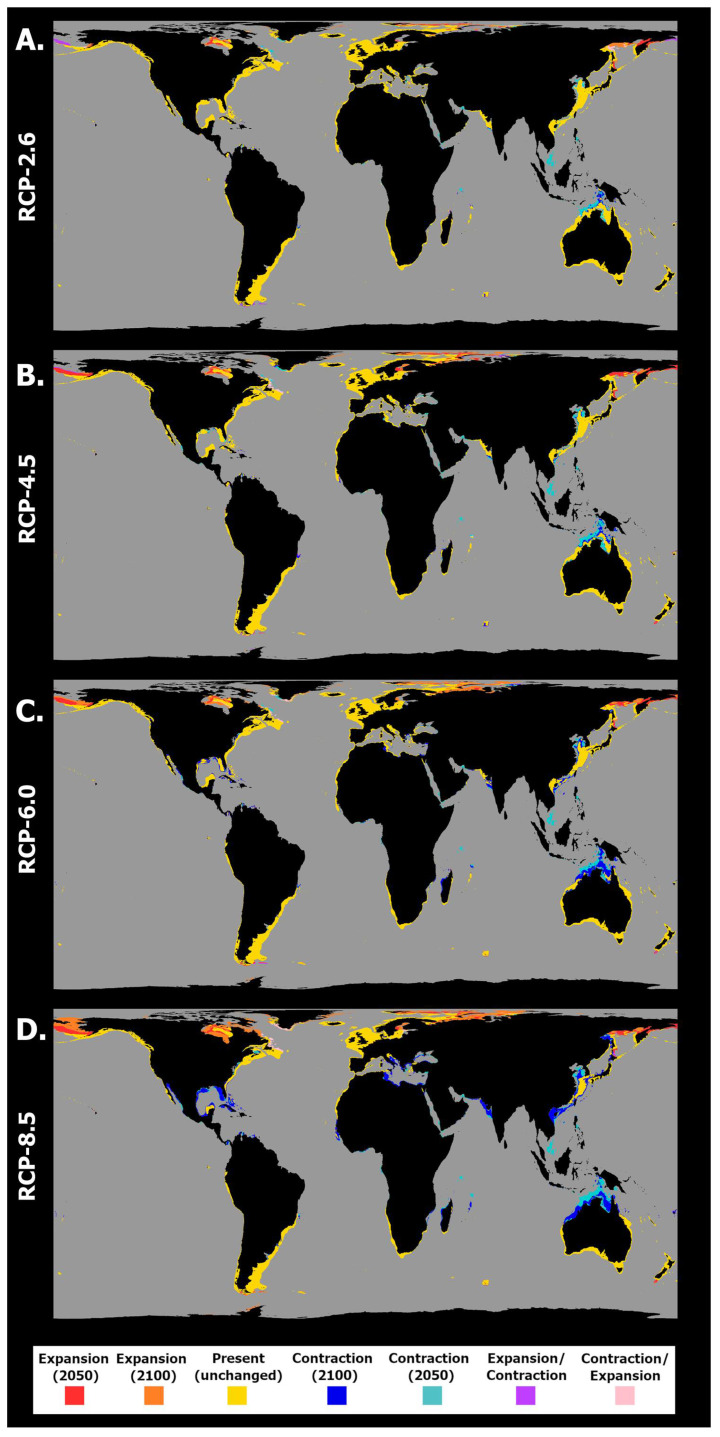
Projected spatiotemporal changes in distribution for *Pseudo-nitzschia seriata* between the present day, 2050, and 2100 across four representative concentration pathway scenarios (**A**) RCP-2.6, (**B**) RCP-4.5, (**C**) RCP-6.0, and (**D**) RCP-8.5. Projected occurrence distribution changes are represented: unidirectional range shifts (i.e., projected expansion in red and orange and projected contractions in dark and light blue) and transitory fluctuations (i.e., range contractions followed by expansions in pink and vice versa in purple).

**Figure 6 toxins-15-00009-f006:**
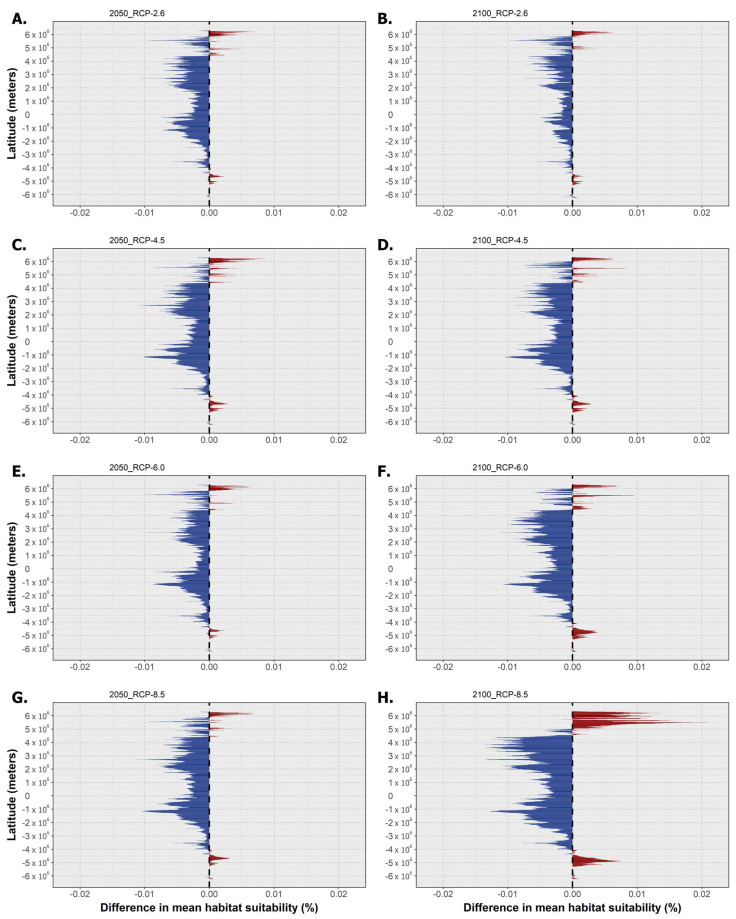
Mean latitudinal habitat suitability temporal changes (i.e., gains in red and losses in blue) between the present day and and 2050 (left—(**A**,**C**,**E**,**G**)) and 2100 (right—(**B**,**D**,**F**,**H**)) for *Pseudo-nitzschia fraudulenta* across the four representative concentration pathway scenarios (RCP-2.6, -4.5, -6.0, and -8.5; CMIP5). The vertical dashed line at ‘0′ represents the present-day baseline.

**Figure 7 toxins-15-00009-f007:**
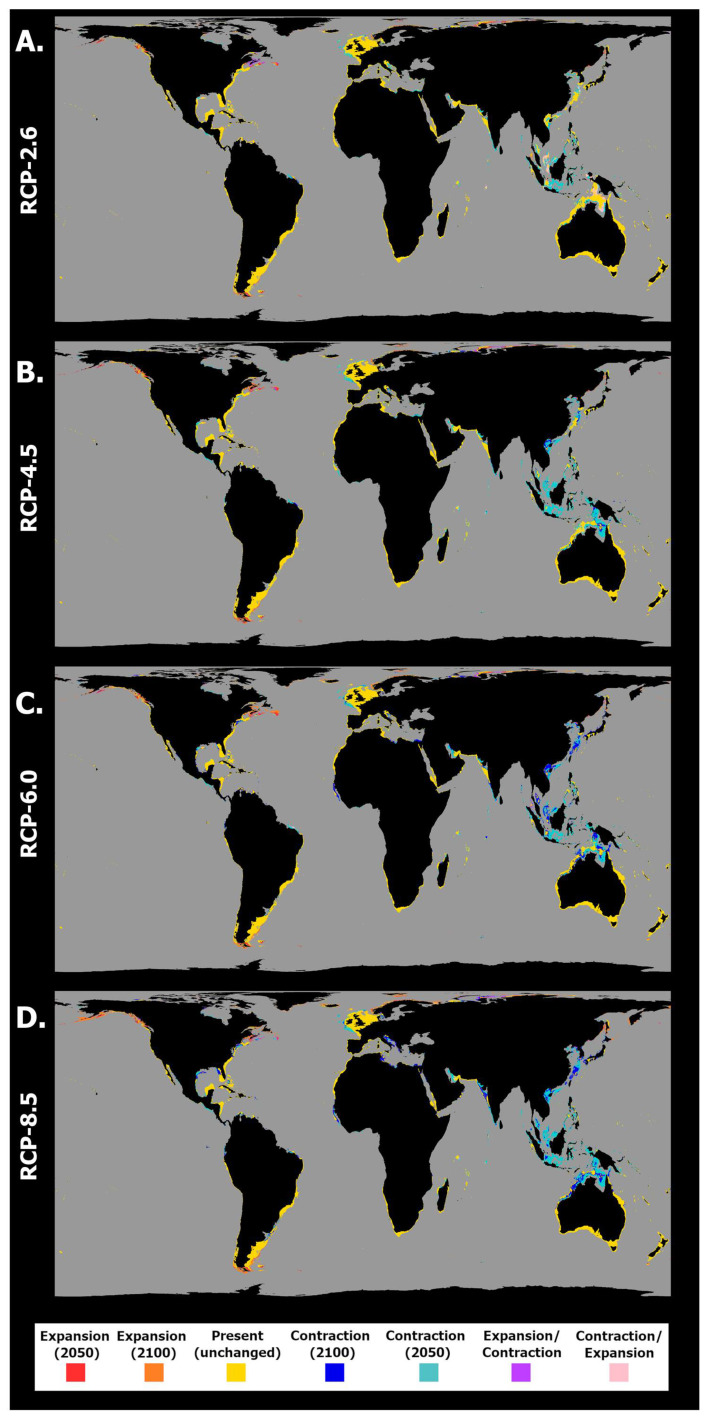
Projected spatiotemporal changes in distribution for *Pseudo-nitzschia fraudulenta* between the present day, 2050, and 2100 across four representative concentration pathway scenarios (**A**) RCP-2.6, (**B**) RCP-4.5, (**C**) RCP-6.0, and (**D**) RCP-8.5. Projected occurrence distribution changes are represented: unidirectional range shifts (i.e., projected expansions in red and orange and projected contractions in dark and light blue) and transitory fluctuations (i.e., range contractions followed by expansions in pink and vice versa in purple).

**Table 1 toxins-15-00009-t001:** Top four most important variables in the ensemble model for each species.

Species	#1	#2	#3	#4
*P. australis*	Bathymetry 71.4%	Temperature Maximum 9.7%	Temperature Range 5.4%	Temperature Minimum 3.4%
*P. seriata*	Bathymetry 81.9%	Temperature Minimum 9.5%	Temperature Range 4.7%	Temperature Mean 0.9%
*P. fraudulenta*	Bathymetry 61.9%	Salinity Minimum 15.2%	Temperature Mean 5.4%	Temperature Maximum 4.6%

**Table 2 toxins-15-00009-t002:** Summary list of the ensemble MaxEnt model predictions for each AST-producing species. Accurate, over-, and underpredictions of distribution based on Trainer et al. [27] and Bates et al. [35].

Species	Accurate prediction	Overprediction	Underprediction
*P. australis*	W Atlantic (Gulf of Mexico, Argentina, Gulf of Maine, South Brazil)	Southern Ocean	Bay of Fundy Gulf of Alaska Bering Strait
	N Atlantic (Celt Sea, Ireland, Scotland, Galicia, Portugal, Morocco)	Arctic Ocean	
	SE Atlantic (Benguela, Namibia)	Indonesia	
	Eastern Pacific (Mexico, Baja California, USA, Alaska, Canada, Peru, Chile)	Indian Ocean (Oman)	
	Bering Strait and Bering Sea	NW Pacific (Okhotsk Sea, Sea of Japan, Eastern China Sea)	
	Oceania (Australia, New Zealand)	Mediterranean Sea	
		Atlantic (northern South America)	
*P. seriata*	NW Pacific (Okhotsk Sea, Sea of Japan, Bering Sea)	Southern Hemisphere	Singapore
	NW Atlantic (Gulf of St. Lawrence, Greenland)		
	Gulf of Mexico		
	Arctic Ocean		
	Black Sea		
*P. fraudulenta*	NW Atlantic (USA, Canada)	Madagascar Indonesia SE Pacific Red Sea Arctic Ocean	Marmara Sea (Turkey)
	NE Atlantic (Morocco, Celt Sea, Scotland)		
	North Sea		
	Mediterranean Sea (NW Mediterranean, Adriatic, Morocco)		
	Gulf of Mexico		
	SW Atlantic (South Brazil, Argentina)		
	NE Pacific (Washington, Gulf of California, Baja California, Mexico, Chile)		
	NW Pacific (Okhotsk Sea, Sea of Japan)		
	Oceania (Australia, New Zealand)		
	Indian Ocean (Pakistan)		

**Table 3 toxins-15-00009-t003:** Precuration, postcuration, and post-environmental-filtering numbers of valid occurrences for each AST-producing species included in the present analysis.

Species	Precuration	Curated	Post-Environmental Filtering
*Pseudo-nitzschia australis* ^A^	57	32	30
*Pseudo-nitzschia seriata* ^B^	1997	834	782
*Pseudo-nitzschia fraudulenta* ^C^	162	129	124

^A.^ Pseudonitzschia australis Frenguelli, 1939 in GBIF Secretariat (2022). GBIF Backbone Taxonomy. Checklist dataset: https://doi.org/10.15468/39omei, accessed via GBIF.org on 17 March 2022. ^B.^ Pseudo-nitzschia seriata (Cleve) H.Peragallo, 1899 in GBIF Secretariat (2022). GBIF Backbone Taxonomy. Checklist dataset: https://doi.org/10.15468/39omei, accessed via GBIF.org on 17 March 2022; ^C.^ Pseudo-nitzschia fraudulenta (Cleve) Hasle, 1993 in GBIF Secretariat (2022). GBIF Backbone Taxonomy. Checklist dataset: https://doi.org/10.15468/39omei, accessed via GBIF.org on 17 March 2022.

## Data Availability

The data are available in the Appendix A.

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
