# Peer review of "Impacts of Climate Change on the Biogeography of Three Amnesic Shellfish Toxin Producing Diatom Species"

_toxins, 2022, doi:10.3390/toxins15010009_

Round 1

Reviewer 1 Report

The evaluation of the geographical extent of Harmful Algal Blooms is a research topic of real importance for the control of ecosystem health and implicitly human health. The authors have developed a particularly extensive and complex scientific work, useful for the academic environment but also for the decision-makers responsible for managing the effects of climate change. Research conceptualization is professional; the aim is properly defined. The research methodology is presented accurately and is structured in well-defined subchapters. The results of the experimental study are also structured in subchapters, being accompanied by relevant, high-quality graphic images. The analysis of the experimental results is fluent, logical, based on pertinent scientific arguments and reliable bibliographic sources. The bibliography studied is relevant and mostly current.

Overall, the manuscript is of very good scientific quality and is carefully and skillfully edited. However, I recommend the authors to do the following minor changes:

1. The word Impacts should be used only once in the title.

2. The Materials and Methods chapter should be placed before the Results and Discussions for a fluent understanding of the text.

3. A Conclusions chapter is required, elaborated briefly based on the main findings; the way of using these results in some concrete actions and strategies should also be mentioned.

Author Response

Response to Reviewer 1

«The evaluation of the geographical extent of Harmful Algal Blooms is a research topic of real importance for the control of ecosystem health and implicitly human health. The authors have developed a particularly extensive and complex scientific work, useful for the academic environment but also for the decision-makers responsible for managing the effects of climate change. Research conceptualization is professional; the aim is properly defined. The research methodology is presented accurately and is structured in well-defined subchapters. The results of the experimental study are also structured in subchapters, being accompanied by relevant, high-quality graphic images. The analysis of the experimental results is fluent, logical, based on pertinent scientific arguments and reliable bibliographic sources. The bibliography studied is relevant and mostly current.»

  • We thank the reviewer for his or her comments and suggestions, and for the time spent in reviewing our manuscript.

«Overall, the manuscript is of very good scientific quality and is carefully and skillfully edited. However, I recommend the authors to do the following minor changes:

The word Impacts should be used only once in the title.»

  • We apologize for the mistake in the original title. This was a leftover from pre-submission. The word Impacts is now featured only once (see Tittle).

«The Materials and Methods chapter should be placed before the Results and Discussions for a fluent understanding of the text.»

       -       While we the authors agree with this view, the journal’s “instructions for authors” states that the manuscript should be structured according to: “Introduction, Results, Discussion, Conclusions (optional), Materials and Methods, Patents”.

«A Conclusions chapter is required, elaborated briefly based on the main findings; the way of using these results in some concrete actions and strategies should also be mentioned.»

  • We have added a Conclusions section at the end of the manuscript (394-404).

Reviewer 2 Report

Thank you for the opportunity to review the paper.

The manuscript is well organized but in some section is a bit confused and understanding is difficult. I think that some steps should be better explained.  I hope that my comments are helpful to the authors in order to improve the quality of their manuscript for acceptance. I try to be brief as much as possible.

The objective of the study was to evaluate the potential effects of future marine climate change on the distribution of three amnesic shellfish toxin (AST) producing  Pseudo-nitzschia species. This study implemented a model to represent the present-day habitat suitability and species distribution, projecting these into two future time periods (i.e., 2050 and 2100) and four RCP scenarios (i.e., RCP-2.6, 4.5, 6.0, and 8.5; CMIP5).

I didn’t find in the text a clear explanation about the RCP scenarios (i.e., RCP-2.6, 4.5, 6.0, and 8.5; CMIP5), which are the basis of the study. I found the definition of “RCP” (Representative Concentration Pathway scenario) only in the caption of the Figures 2, 3, 4, but never in the text. Moreover I didn’t find in the text a clear explanation of the meaning of the 2.6, 4.5, 6.0 scenario as well as of CMIP5.

In addition, more details regard the data collected and about the Global Biodiversity Facility database are necessary.

In my humble opinion, more details are needed to better understand all the study carried out.

Below you can find few more timely comments.

Line 60: …where constitutes potential poleward….

Line 74 …ranges across time and/or spaces..

Line 89:..spatio-temporal trends..

Line 415: uncompleted sentence

Author Response

Response to Reviewer 2

«Thank you for the opportunity to review the paper.

The manuscript is well organized but in some section is a bit confused and understanding is difficult. I think that some steps should be better explained.  I hope that my comments are helpful to the authors in order to improve the quality of their manuscript for acceptance. I try to be brief as much as possible.

The objective of the study was to evaluate the potential effects of future marine climate change on the distribution of three amnesic shellfish toxin (AST) producing Pseudo-nitzschia species. This study implemented a model to represent the present-day habitat suitability and species distribution, projecting these into two future time periods (i.e., 2050 and 2100) and four RCP scenarios (i.e., RCP-2.6, 4.5, 6.0, and 8.5; CMIP5).»

  • We thank the reviewer for his or her comments and suggestions, and for the time spent in reviewing our manuscript.

«I didn’t find in the text a clear explanation about the RCP scenarios (i.e., RCP-2.6, 4.5, 6.0, and 8.5; CMIP5), which are the basis of the study. I found the definition of “RCP” (Representative Concentration Pathway scenario) only in the caption of the Figures 2, 3, 4, but never in the text. Moreover I didn’t find in the text a clear explanation of the meaning of the 2.6, 4.5, 6.0 scenario as well as of CMIP5.»

  • A short paragraph has been added in the Material and Methods section (L443-453) describing the RCP scenarios in terms of emission/temperature increase trends and the CMIP5. We have also added a definition of RCP in the introduction, as it was lacking (L88-89).

«In addition, more details regard the data collected and about the Global Biodiversity Facility database are necessary.»

  • We have added a short sentence on the GBIF database (L411-412). We have also added the citation from GBIF for each species in the reference list.

Pseudo-nitzschia fraudulenta (Cleve) Hasle, 1993 in GBIF Secretariat (2022). GBIF Backbone Taxonomy. Checklist dataset https://doi.org/10.15468/39omei accessed via GBIF.org on 2022-03-17.

Pseudo-nitzschia seriata (Cleve) H.Peragallo, 1899 in GBIF Secretariat (2022). GBIF Backbone Taxonomy. Checklist dataset https://doi.org/10.15468/39omei accessed via GBIF.org on 2022-03-17.

Pseudonitzschia australis Frenguelli, 1939 in GBIF Secretariat (2022). GBIF Backbone Taxonomy. Checklist dataset https://doi.org/10.15468/39omei accessed via GBIF.org on 2022-03-17.

«Line 60: …where constitutes potential poleward….»

  • We have changed the sentence section accordingly (L60).

«Line 74 …ranges across time and/or spaces..»

  • We have changed the sentence accordingly (L74).

«Line 89:..spatio-temporal trends..»

  • We have changed the sentence accordingly (L90).

«Line 415: uncompleted sentence»

  • We have completed the sentence in question (L428-430).

Reviewer 3 Report

The manuscript “Impacts of Climate Change Impacts on the biogeography of three Amnesic Shellfish Toxin-producing Diatom species” describes the potential distribution changes on these three species based on species distribution models based on temperature, salinity, current velocity and bathymetry under different IPCC based scenarios. This is an important topic, I do however have some serious concerns about this study and can therefore not recommend publication unless the authors are able to make significant changes to their model.

My biggest concern is the choice of the abiotic predictor variables used. With the exception of temperature none of the four variables will have a direct significant impact on phytoplankton growth/abundance. Bathymetry was found to be the most important variable but this is most likely only the case because bathymetry has an impact on upwelling and tidal mixing in coastal regions so it will impact light and nutrient conditions. However, it is important to understand that two different coastal regions with the same bathymetry will NOT have the same light and nutrient conditions as this depends on many other factors (e.g. upwelling region or not), so using bathymetry as a predictor variable is not suitable. At the same time the two most critical abiotic factor that directly influence phytoplankton growth in general have been completely ignored (light and nutrient availability/ nutrient ratios). To make matters more complicated, both light and nutrient conditions are significantly impacted by mixing depth which in turn is affected by temperature. So this needs to be implemented into the model and thoroughly discussed.

Based on table 2 the model used does not seem to work very well. Especially the overprediction of P. australis in so many very different regions is clearly a sign to me that the model isn’t a good tool to predict possible future changes.  

In line 318-319 you mention that Pseudo-nitzschia blooms are linked to upwelling regions. This is correct and is thought to be related to the high nutrient concentrations in these areas, but nutrient concentrations are not included in your model nor is the importance of nutrients even discussed here. I assume nutrient concentrations and especially the ratio of different nutrients is one of the reasons why your model is over- or underestimating the abundance of different Pseudo-nitzschia species in so many regions.

Of course ocean warming is likely to have a direct impact on species distribution but it will also change stratification patterns in coastal and open ocean regions and might therefore also indirectly affect species because of changes in the nutrient and light conditions. These factor need to be taken into account, especially since it is well known that Pseudo-nitzschia blooms are strongly linked to nutrient availability (Palma et al 2010, Schnetzer et al 2013, Torres Palenzuela et al 2019).

Palma et al. 2010. Can Pseudo-nitzschia blooms be modeled by coastal upwelling in Lisbon Bay? Harmful Algae, 9, 294303, doi:10.1016/j.hal.2009.11.006.

Schnetzer et al. 2013. Coastal upwelling linked to toxic Pseudo-nitzschia australis blooms in Los Angeles coastal waters, 20052007. Journal of Plankton Research, 35, 10801092, doi:10.1093/plankt/fbt051.

Torres Palenzuela, et al. 2019. Pseudo-nitzschia blooms in a coastal upwelling system: Remote sensing detection, toxicity and environmental variables. Water, 11, 1954, doi:10.3390/w11091954.

Author Response

Response to Reviewer 3

«The manuscript “Impacts of Climate Change Impacts on the biogeography of three Amnesic Shellfish Toxin-producing Diatom species” describes the potential distribution changes on these three species based on species distribution models based on temperature, salinity, current velocity and bathymetry under different IPCC based scenarios. This is an important topic, I do however have some serious concerns about this study and can therefore not recommend publication unless the authors are able to make significant changes to their model.»

  • We thank the reviewer for his or her comments and suggestions, and for the time spent in reviewing our manuscript.

«My biggest concern is the choice of the abiotic predictor variables used. With the exception of temperature none of the four variables will have a direct significant impact on phytoplankton growth/abundance. Bathymetry was found to be the most important variable but this is most likely only the case because bathymetry has an impact on upwelling and tidal mixing in coastal regions so it will impact light and nutrient conditions. However, it is important to understand that two different coastal regions with the same bathymetry will NOT have the same light and nutrient conditions as this depends on many other factors (e.g. upwelling region or not), so using bathymetry as a predictor variable is not suitable. At the same time the two most critical abiotic factor that directly influence phytoplankton growth in general have been completely ignored (light and nutrient availability/ nutrient ratios). To make matters more complicated, both light and nutrient conditions are significantly impacted by mixing depth which in turn is affected by temperature. So this needs to be implemented into the model and thoroughly discussed.»

«Based on table 2 the model used does not seem to work very well. Especially the overprediction of P. australis in so many very different regions is clearly a sign to me that the model isn’t a good tool to predict possible future changes. »

«In line 318-319 you mention that Pseudo-nitzschia blooms are linked to upwelling regions. This is correct and is thought to be related to the high nutrient concentrations in these areas, but nutrient concentrations are not included in your model nor is the importance of nutrients even discussed here. I assume nutrient concentrations and especially the ratio of different nutrients is one of the reasons why your model is over- or underestimating the abundance of different Pseudo-nitzschia species in so many regions.

«Of course ocean warming is likely to have a direct impact on species distribution but it will also change stratification patterns in coastal and open ocean regions and might therefore also indirectly affect species because of changes in the nutrient and light conditions. These factor need to be taken into account, especially since it is well known that Pseudo-nitzschia blooms are strongly linked to nutrient availability (Palma et al 2010, Schnetzer et al 2013, Torres Palenzuela et al 2019).

 Palma et al. 2010. Can Pseudo-nitzschia blooms be modeled by coastal upwelling in Lisbon Bay? Harmful Algae, 9, 294303, doi:10.1016/j.hal.2009.11.006.

Schnetzer et al. 2013. Coastal upwelling linked to toxic Pseudo-nitzschia australis blooms in Los Angeles coastal waters, 20052007. Journal of Plankton Research, 35, 10801092, doi:10.1093/plankt/fbt051.

Torres Palenzuela, et al. 2019. Pseudo-nitzschia blooms in a coastal upwelling system: Remote sensing detection, toxicity and environmental variables. Water, 11, 1954, doi:10.3390/w11091954.»

  • Although we understand and agree that more predictor variables would be beneficial to the projections, namely the inclusion of light and nutrient conditions, unfortunately (and to the best of our knowledge) there are currently no readily available projection layers for these two predictors for the future and at a global scale. Bio-Oracle and MARSPEC, two of the most well-known downscalled marine layer databases, do not provide these. Indeed, this is one of the major limitations to marine SDMs at the moment, as models are still very limited in terms of predictor choice. To address this issue, we have decided to revisit the caveats and limitations section of the manuscript (L366-374) in order to adequately discuss this, expanding on the unaccounted predictor variables part (see below).

«In the present manuscript, due to the inexistence of readily available, downscaled projec-tions of environmental predictors such as nutrient ratios and light, for the future and for each of the RCP scenarios employed, presented a limitation to the ecological relevance of the models employed. Nutrient and light conditions are of paramount importance for shaping marine phytoplankton abundance and distribution, particularly for Pseu-do-nitzschia spp. blooms (Palma et al., 2010; Schnetzer et al., 2013; Torres Palenzuela et al., 2019) In this sense, ecological modelling of future marine climate change biodiversity im-pacts requires the expansion of already existing environmental and abiotic layer online databases (such as Bio-ORACLE, MARSPEC, etc.) so that these types of layers are readily available, and at global scales whenever possible.»

  • Concerning overprediction, this is discussed in the same section as a limitation. Nonetheless, the overall trends of potential poleward shifts are consistent with the literature and with the projected changes for other similar species.

Round 2

Reviewer 2 Report

Thanks to the authors for accepting my suggestions. In my humble opinion the understanding of the manuscript is clearer and the paper can be accepted in present form

Reviewer 3 Report

I am happy with the way the authors have addressed my concerns in the discussion section